# Phytotherapy-Induced Hepatocytotoxicity: A Case Report

Stephen Malnick [1,2] , Ali Abdullah [1,2], Yaacov Maor [2] and Manuela G. Neuman [3,*]

1 Department of Internal Medicine C, Kaplan Medical Center, Hebrew University, Rehovot 76100, Israel; steve@stevemalnickmd.com (S.M.); onco82@hotmail.com (A.A.)
2 Institute of Gastroenterology and Liver Disease, Kaplan Medical Center, Rehovot 76100, Israel; halishy@netvision.net.il
3 In Vitro Drug Safety and Biotechnology and Department of Pharmacology and Toxicology, University of Toronto, Toronto, ON M5G OA3, Canada
* Correspondence: m_neuman@rogers.com

**Abstract:** Herbal and complementary medicine are frequently integrated with conventional medicine. We aim to report a case of severe herbal-induced liver injury (HILI) due to chronic use of green tea and protein shake. We present both clinical and laboratory evidence implicating mitochondrial toxicity and an immune response leading to a hypersensitivity reaction to the products. We have recently treated a 39-year-old man with hepatotoxicity resulting from a combination of a green tea-containing powder and a branched-chain amino acid supplement that was commenced 2 months previously. The hepatotoxicity resolved by stopping the consumption of these products and no other cause was detected. We decided to perform a lymphocyte toxicity assay (LTA) to determine if there was laboratory support for this diagnosis. LTA (% toxicity) represents the response of the mitochondria to toxic injury. To determine the role of the proinflammatory and anti-inflammatory cytokines and chemokines in the patient's reaction, we measured the level of cytokines and chemokine in the media of growing cells, exposed to each product or to a combination of products. The increased cytokines and chemokines are presented as the x-fold elevations from the upper limit of normal (ULN) for matrix metalloproteinase (MMP) (pg/mL × 1.5 ULN) and interleukin (IL)-1β (pg/mL × 1.8 ULN). Higher elevations were found for interferon (IFN)-β, IFN-γ, IL-8, IL 13, IL-15 (pg/mL × 2 ULN), regulated upon activation, normal T cell expressed and presumably secreted (RANTES) (pg/mL × 2 ULN), and nuclear factor (NFκB) (pg/mL × 3 ULN). The highest increases were for vascular endothelial factor (VEGF) (pg/mL × 10 ULN), tumor necrosis factor (TNF)-α, and tumor necrosis factor-related apoptosis-inducing ligand (TRAIL) (pg/mL × 13 ULN). An examination of cellular markers showed the difference between programmed cell death (apoptosis) and cell death due to necrosis. In our case, cytokeratin—ccK18 (M-30) U/L was within the normal limits, suggesting that apoptosis was normal, while ccK8(M65) U/L was elevated at 1.5 × ULN. This result implies that upon the treatment of the patient's lymphocytes with the products, the mechanism of toxicity is necrosis. In susceptible individuals, the combination of protein and herbal tea produces mitochondrial toxicity and a strong T-lymphocyte-1 response, leading to HILI. There is a need of international reporting of adverse drug reactions by clinicians, laboratories, and pharmaceutical manufacturers to drug regulatory authorities. This requires internationally accepted standard definitions of reactions, as well as criteria for assessment.

**Keywords:** drug-induced liver injury; herbal-induced liver injury; lymphocyte toxicity assay; proinflammatory cytokines; caspase; apoptosis; necrosis

## 1. Introduction

Drug-induced liver injury (DILI) causes significant morbidity and mortality [1]. Hepatotoxicity caused by exposure to a drug or an herbal compound results in injury or damage to the liver that may be associated with impaired liver function [2,3]. This hepatotoxicity

may result in liver failure, especially if the administration of the offending agent continues [3]. Type I lesions are "predictable", dose- and time-dependent, occurring in individuals exposed to appropriate doses of the causative therapeutics. The predictable lesions are reproducible in animal studies. Type II lesions are those that are "unpredictable", dose- and time-independent, occurring only in susceptible individuals exposed to appropriate doses. The lesions are not reproducible in animals.

Drug-induced hepatotoxicity is one of the most common causes of termination of drug development and an important cause of the withdrawal of market authorization for a product [4]. Hepatotoxicity may result from the direct action of the drug, herbal medicine, and venom, or indirectly, from interactions with other foods or xenobiotics [5–9]. Genetic polymorphisms may affect the reactive metabolite at the cellular level or trigger damage directly [10–12]. Toxicity may also be increased by the presence of another medical condition that impairs liver function [13,14].

Prescription medicines are thoroughly tested prior to receiving regulatory approval but despite this, real-world experience reveals toxicity that was undetected in clinical trials [15–18]. In addition to intrinsic drug reactions, there are also idiosyncratic (iDILI) reactions in susceptible individuals exposed to therapeutic doses [19].

Herbal preparations are present in many complementary and alternative medicines worldwide. Consumers often consider these products harmless, since they are "natural" foods. In the USA, up to 42% of people consume such herbal medications and up to one third of patients in liver clinics in the United States report the use of herbal agents. These products are regarded as food supplements, and their use in not regulated or controlled.

One of the most common adverse reactions to herbal preparations is toxic hepatitis [9]. In many cases, the evidence is based on case reports, with a lack of laboratory evidence to define a possible mechanism.

Tea, alone or in combination with other ingredients used for weight loss, has been shown to result in hepatocytotoxicity [20–24].

We have seen a 39-year-old man with hepatotoxicity resulting from a combination of a green tea-containing powder and a branched chain amino acid supplement. We present evidence suggesting a profound stimulation of proinflammatory cytokine production in this patient.

## 2. Case Report

A previously healthy 39-year-old man was referred for investigation of elevated liver enzymes. He was previously healthy and had no regular medications. He denied excessive consumption of alcohol, sweet sugary beverages, and diet sodas. Two months previously, he had started to consume daily a combination of a protein powder containing green tea and branched chain amino acids (BCAA), Lean Energy, from EVL nutrition (https://www.evlnutrition.com/, accessed on 11 November 2023), together with Alfa Whey protein powder, manufactured by Sommer Laboratories Ltd., Rosh Ha'ayin, Israel.

Physical examination was unremarkable. Abdominal ultrasound was unremarkable. Liver enzymes were in the normal range on routine blood tests performed 2 years previously. Results of serum ceruloplasmin, alpha-1-antitrypsin, celiac serology, autoimmune antibodies were all negative. There was no evidence for viral hepatitis B or C infection. Viral hepatitis A IgG was positive, and IgM was negative. Human herpes virus (HHV) 6 and 7 were negative.

The results of ALT, AST, and GGT are shown in Table 1. The hepatic synthetic function, as assessed by INR and serum albumin, was normal.

In view of the history of consumption of both green tea and branched chain amino acid supplements, we recommended to cease consuming these products.

His liver enzymes slowly declined to normal levels. The patient declined liver biopsy and to be rechallenged with the same product. We suggested to him to performing the lymphocyte toxicity assay (LTA). The lymphocyte reaction to one or a combination of the two consumed products will determine their implication in the patient's illness.

The patient agreed to LTA and to all immune analysis.

**Table 1.** Dynamic of liver enzymes and bilirubin in our patient.

| Date | ALP | ALT | AST | GGT | Bilirubin Total/Direct |
|---|---|---|---|---|---|
| Normal range | 30–280 IU/L | 0–45 IU/L | 0–35 IU/L | 0–55 IU/L | 0.3–1.2 mg/dL 0.1–0.3 mg/dL |
| 27 February 2023 | 90 | 127 | 66 | 17 | 0.4/0.1 |
| 9 March 2023 | 87 | 202 | 119 | 20 | 0.4/0.1 |
| 27 April 2023 | 77 | 67 | 44 | 18 | 0.4/0.1 |
| 30 July 2023 | 91 | 43 | 35 | 16 | 0.4/0.1 |
| 1 September 2023 | 90 | 32 | 35 | 20 | 0.4/0.1 |
| 16 November 2023 | 91 | 41 | 35 | 21 | 0.4/012 |

## 3. Methods

We performed studies to determine if tea and protein, given together or alone, may be responsible for the toxic reaction(s) involving mitochondrial succinate dehydrogenase (SDH). Immunologic and inflammatory pathways have also been examined.

We performed in vitro challenge of the patient's lymphocytes with the herbal product. The lymphocytes were extracted following the centrifugation of fresh blood obtained from the patient, using a density gradient. The lymphocytes were incubated with therapeutic concentrations of the herbal products (tea, protein, and both tea and protein together) in the presence or absence of a metabolizing system. Following a 24 h incubation of the cells with the products, the supernatant (in which the lymphocytes were cultivated) was removed, and the lymphocytes were further incubated with a yellow tetrazolium dye, 3-(4,5-dimethylthiazol-2-yl)-2,5-diphenyl—tetrazolium bromide (MTT)). The degree of MTT reduction to the water-insoluble purple formazan represents a measure of succinate dehydrogenase (SDH) activity, which in turn indicates cellular viability [8]. Toxicity to the natural remedies alone or in combination was expressed as the percentage of reduction in dye intensity relative to the control. The toxic effect was presented by the percentage of the patient's cells that died during the 24 h exposure in the presence of a metabolizing system and tea, protein, or their combination. The results were corrected for toxicity due to each of the products (in the absence of a metabolizing system). A lymphocyte toxicity assay was performed similarly during both the acute HILI episode and several years later, due to the presence of memory cells [8].

The innate immune system also participates in a hypersensitivity reaction. Hepatocyte necrosis activates the innate system, most likely through proinflammatory mediators (e.g., cytokines, chemokines). These mediators can be directly cytotoxic, or they can lead to recruitment of cells of the innate immune system [8,25]. The innate immune system may also have a role in recovery via anti-inflammatory components. Cytokines play an essential role in the inflamed liver [8,25,26]. To analyze the mechanism of liver damage, we examined a panel of cytokines. The levels of cytokines were quantitatively determined in serum, as well as in the supernatant in which the lymphocytes grew either in the presence or in the absence of the herbal remedies. The cytokine determinations were performed using enzyme-linked immunosorbent-assay (ELISA) [8].

The cytokines and chemokines determined were as follows: IL-2, IL-4, IL-15, MMP, TGF-β, RANTES, TRAIL, TNF-α (R&D Systems, Inc., Minneapolis, MN, USA), IL-1, IL-6, IL-8, VEGF (PeproTech, Cranbury, NJ, USA), NF–κp65 -Total/Phospho InstantOne™ (eBioscience, San Diego, CA, USA), and IFN kits (InVitrogen Corporation, Camarillo, CA, USA). The determinations were performed according to the manufacturers' specifications.

Aliquots from the sample were added to the 96-well plate. Utilizing ELISA enables the acquisition of high-quality precision and quantification data for desired markers in a biological sample. Each specimen was analyzed in duplicate. Our measurement system demonstrates strong correlations across replicates, with correlation coefficients >0.99,

ensuring reliable detection of differences in cytokine levels between biological samples. The intra-assay coefficient of variance (CV), which measures the variance between data points within an assay (meaning that the sample replicates ran within the same plate), was 1.5–2% [8].

Aliquots from the sample were added to the 96-well plate. Each specimen was analyzed in duplicate, with 95% sensitivity and 92% specificity. Our measurement system demonstrated strong correlations across the replicates, with correlation coefficients >0.99, ensuring reliable detection of differences in cytokine levels between biological samples. Intra-assay CV, which is a measure of the variance between data points within an assay (meaning that the sample replicates ran within the same plate), was 1.5–2% [8]. To understand the toxic effect of the product, we clarified the mechanism leading to cell death.

*Apoptosis* is triggered by host conditions, including starvation, a stressful environment, and the presence of toxins [25]. To confirm the mechanism, we chose to analyze apopto-necrotic biomarkers.

These markers can also determine if cell death was caused by necrosis or by programmed cell elimination—apoptosis. The death ligand tumor necrosis factor-related apoptosis-inducing ligand (TRAIL), a member of the TNF cytokine superfamily, has long standing apoptotic activity against normal cells.

Cytokeratin 18 encodes the type I, intermediate filament chain keratin 18. Keratin 18 and its filament partner, keratin 8, are expressed in single-layer epithelial tissues.

We measured the cytokeratins in sera and media of the cultured lymphocytes using M30 and M65 antibodies. M30 is specific for apoptosis and M65 combines death processes from both apoptosis and necrosis. In epithelial cells, one of those substrates is the intermediate filament protein keratin 18 (K18). The M30 antibody recognizes a neo-epitope exposed after the caspase cleavage of K18 after the aspartic acid residue.

Cleavage at this position occurs early during apoptosis by caspase 9 and during the execution phase by caspase 3 and caspase 7. The M30 Apoptosense® ELISA measures the levels of soluble caspase-cleaved K18 (ccK18) fragments containing the K18Asp396 neo-epitope [5]. The ccK18 level increases during apoptosis and is inhibited by the caspase-inhibitor zVAD-FMK. M65® ELISA measures total K18. Combining the two assays, M30 and M65, is useful for the assessment of cell death by both apoptosis and necrosis. The markers cleaved cytokeratins CK 18 and CK 8 (M30 and M65) were quantified using kits from Bender MedSystems (Vienna, Austria).

The correlation coefficient was linear ($r = 0.990$). NFκB is predominantly localized in cytoplasm as a mediator of cell survival. TRAIL belongs to the TNF super-family and shares a very high amino acid identity. TRAIL is an important component of the granule-independent cytolytic mechanism of cytotoxic lymphocytes [27].

We calculated the area under the curve (AUC) of these components. The area under the plot of plasma concentration of a drug versus time after dosage AUC gives insight into the extent of exposure to a drug and its clearance rate from the body. By integrating over time rather than looking at individual concentration measurements, a more accurate estimate of the overall exposure to the drug is obtained. Such measurements have also been found to be meaningful for assessing the net pharmacologic response to a given dose of a drug. These markers had an AUROC of 0.90, with 80% sensitivity and 90% specificity for detecting cell death.

We used standard and reference reagents available from the National Institute for Biological Standards and Controls (NIBSC, Herts, UK). These methods were standardized in our laboratory according to the procedures described.

The RealStar® HHV-6 PCR Kit 1.0 is an in vitro diagnostic test based on real-time PCR technology for the detection, differentiation, and quantification of human herpes virus 6A (HHV-6A) and human herpes virus 6B (HHV-6B), as well as HHV-7-specific DNA. We used the RealStar HHV-6 PCR Kit 1.0, Altona Diagnostic GmbH, Hamburg, Germany, and the real-time PCR instrument Rotor-Gene™ 3000/6000 (Corbett Research, Mortlake, NSW, Australia) for our determinations [28].

Figure 1 presents the lymphocyte toxicity assay in our patient and in a control individual that was not known to be sensitive to tea and protein. Both individuals consented to the analysis.

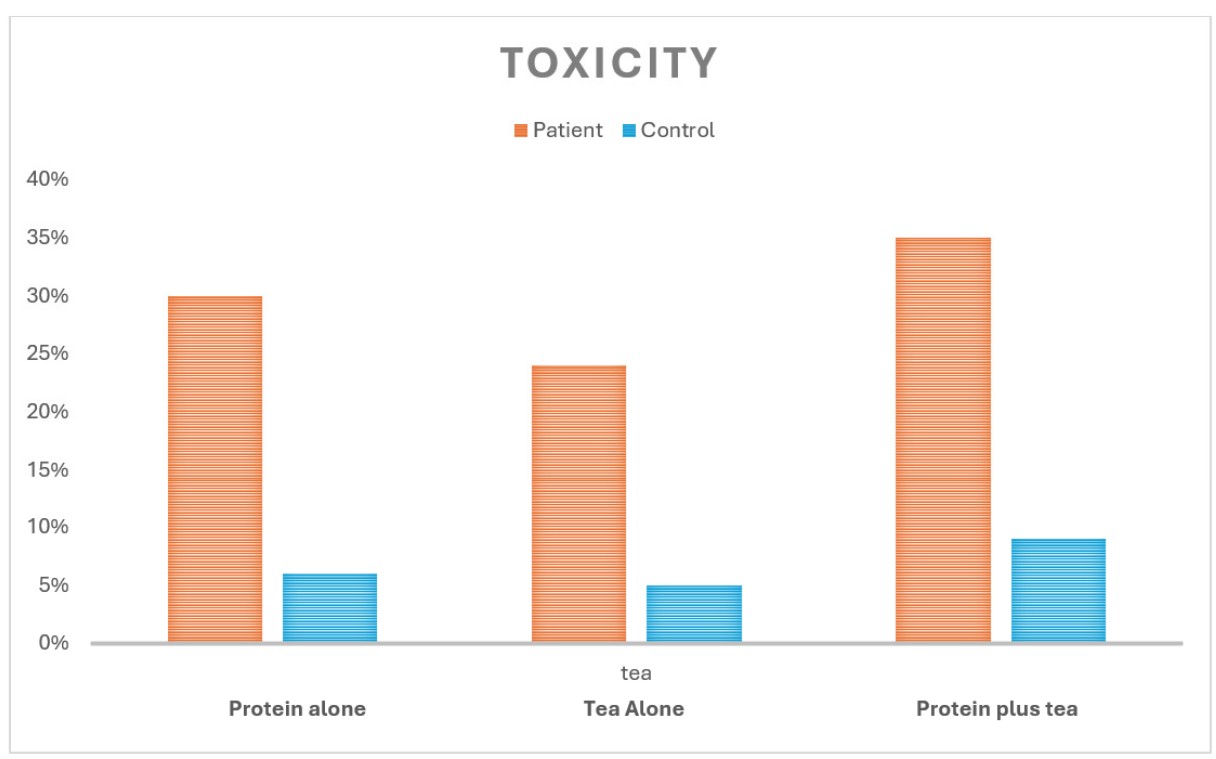

Patient toxicity (yellow)          Control toxicity (blue)

| | | | |
|---|---|---|---|
| I- | protein alone | 30% | 6% |
| II- | tea alone | 24% | 5% |
| III- | protein+ tea | 35% | 9% |

Analysis was performed September 2023.

**Figure 1.** LTA (% toxicity).

The blood was taken in a heparin-coated tube and kept at room temperature. It arrived in Toronto, at the In Vitro Drug Safety and Biotechnology Laboratory, 17 h after the blood was taken from the patient in Israel.

### 3.1. Pathology

No histopathology was available for the patient, since liver biopsy was not performed.

### 3.2. Cytokines and Chemokines

The proinflammatory cytokines and chemokine in serum were elevated in the same blood sample used for LTA. Interleukins, IL-2, IL-4, and IL-6 (pg/mL) were within normal values, while IL-1β (pg/mL) was at 1.8 × the upper limit of normal (ULN), and IL-8, IL 13, and IL-15 were elevated by 2.0 × the ULN. The matrix metalloproteinase (MMP, pg/mL) elevation was 1.5 × the ULN.

IFN-β, IFN-γ, and RANTES (pg/mL) were 2 × the ULN; NFκβ pg/mL was 3 × the ULN; VEGF (pg/mL) was 10 × the ULN. TNF-α and TRAIL (pg/mL) were 13 × the ULN. Our cellular markers showed a difference between programmed cell death (apoptosis) and cell death due to necrosis. In our case, the apoptosis marker, ccK18 (M-30) U/L, is within the normal limits, showing that the cellular death by apoptosis was within the normal limits, while ccK8 (M65) U/L was 1.5 × the ULN. That means that the cells died mostly by necrosis (Table 2).

**Table 2.** Serum levels of cytokines, chemokines, apoptosis, and necrosis markers in the patient.

| Cytokine | Normal Values | Sera-09-1-2023 | Medium 24 h Exposure |
|---|---|---|---|
| IFN-β pg/mL | 20–40 | 44 | 22 |
| IFN-γ pg/mL | 15–40 | 80 | 20 |
| IL-1beta pg/mL | 20–50 | 90 | 35 |
| IL-2 pg/mL | 15–30 | 30 | 36 |
| IL-4 pg/mL | 15–30 | 20 | 18 |
| IL-6 pg/mL | 30–60 | 52 | 54 |
| IL-8 pg/mL | 20–60 | 120 | 42 |
| IL 13 pg/mL | 24–40 | 80 | 44 |
| IL-15 pg/mL | 15–30 | 60 | 38 |
| MMP pg/mL | 20–40 | 60 | 52 |
| NFκB pg/mL | 15–40 | 126 | 46 |
| RANTES pg/mL | 15–50 | 46 | 50 |
| TGF-β ng/mL | 20–40 | 40 | 32 |
| TNF-α pg/mL | 25–50 | 650 | 66 |
| TRAIL pg/mL | 15–40 | 620 | 40 |
| VEGF pg/mL | 20–90 | 906 | 85 |
| ccK18 (M-30) U/L | 68–132 | 130 | 80 |
| ccK8 (M65) U/L | 62–200 | 300 | 80 |

Vascular endothelial growth factor (VEGF) was markedly elevated. This shows that the inflammatory process, resulting in high levels of TNF and TRAIL, negatively influenced the vascularization of the liver. The mitochondrial markers M30 and M65 revealed a predominant level of necrosis compared to apoptosis. The severe HILI resulting from the protein and herbal tea is consistent with an inflammatory picture. This is the first report of the cytokine disturbances associated with HILI from the combination of tea and shake protein (BCAA+ Alpha Whey). Moreover, this is a clear demonstration of hypersensitivity-induced lymphocyte death linked to the same product combination.

## 4. Discussion

Concomitant with the increasing use of herbal products has been an increase in the reports of serious adverse effects, including hepatotoxicity. We report here a case of prolonged hepatotoxicity secondary to consumption of green tea and protein powder that is likely to be related to the chronic administration of these two products: BCAA Lean Energy (EVL Nutrition), together with Alfa Whey protein powder (Sommer Laboratories Ltd.).

The Naranjo score was 5, making this a probable drug reaction. The Naranjo score has been developed for drug toxicity whereas other scoring systems are recommended for the assessment of HILI.

The CIOMS score was the scoring system of choice. The Council for International Organizations of Medical Sciences (CIOMS) undertook a pilot project to establish criteria and proposed its adoption through expert consensus meetings within the official French network of pharmacovigilance. Under CIOMS auspices, an international meeting was organized to test the feasibility of adapting the outcomes from the French consensus meetings for international use in assessing drug-induced liver disorders. The meeting resulted in a series of proposed standard designations for drug-induced liver disorders and criteria of causality assessment [27].

The CIOMS score was developed following consultation with an international expert panel and using data obtained from cases with re-exposure. In this case of toxicity related to the products, the CIOMS score was 9, and this score is also consistent with a probable case of herbal hepatotoxicity. For ethical reasons, we did not perform rechallenge, but there was prolonged rechallenge by the patient since he did not associate the herbal products with anything potentially hepatotoxic. We also considered Hy's Law to estimate severity and the likelihood that the product is causing severe hepatotoxicity. Hy's Law is based on

the combined evidence of hepatic injury, decreased hepatic function, and the absence of other disease-induced damage [3]. Hy's Law requires three criteria to be met: elevation of ALT or AST activity >3 × ULN (indicative of injury); TB function: >2 × ULN (indicative of function); TB and clinical verification to ensure that the effect is therapeutic or health product-induced rather than disease or another cause of injury. The patient we described met the criteria. Moreover, we demonstrated laboratory evidence of a positive lymphocyte toxicity assay [5].

Hepatotoxicity by herbs is defined as an elevation of ALT (alanine aminotransferase) more than five times the upper limit of normality (ULN) or ALP (alkaline phosphatase) more than two times the ULN. In the presence of symptoms, a value greater than three times the ULN of ALT is sufficient. A chronology between the use of the suspected substance and compatible liver damage is required, in addition to the exclusion of alternative causes. These elements can be quantified through a scale [the Roussel Uclaf Causality Assessment Method (RUCAM)] that can be applied as a diagnostic tool [28]. This case report is unique in that there are robust laboratory data suggesting a profound immune stimulation in this patient that led to a proinflammatory cytokine storm, resulting in hepatotoxicity. The response to the combination of both the tea and the protein powder resulted in a stronger mitochondrial response to the toxic substances. The fact that in vitro rechallenge via the lymphocyte toxicity assay months after the cessation of immune modulatory therapy still resulted in the stimulation of the lymphocytes strongly supports an idiosyncratic reaction. Mitochondrial toxicity in this patient was consistent with most of the published cases being linked to this combination. Moreover, the high levels of TNF revealed that apoptosis occurred via caspase-dependent pathways, since both caspase 8 and effector caspase 3 can be activated. The specific mitochondrial enzyme succinate dehydrogenase activity (SDH) showed a clear contribution of mitochondria in the toxicity produced by the herbal products in this patient.

All the proinflammatory cytokines and chemokines levels in the sera were extremely elevated, indicating the involvement of an immune response in the hepatotoxic reaction.

The elevated VEGF indicated that the combination of products led to a vascular change in the liver. Other cytokines, such as MMP and TGF contribute to shifting the ECM in the liver [29]. In the sera of our patient, the levels of these proteins were in a normal range, indicating that there was little or no change in the fibrinogenic pathway.

The significance of this study lies in its comprehensive assessment of clinical and laboratory methods, as well as a combined performance and analysis ranking. This is of value to clinicians confronted with multiple testing options, who may be uncertain about what tests are most useful and what thresholds perform well.

As part of the innate immune system, the Toll-like receptor (TLR)-signaling pathway contributes to the first line of defense against microbial pathogens. The innate immune system was historically considered nonspecific in its response to different invading pathogens, targeting a wide array of organisms, including viruses and bacteria.

Different TLRs recognize specific pathogen-associated molecular patterns (PAMPs). The chemical nature of these PAMPs is highly diverse. For instance, lipopolysaccharide (LPS) of Gram-negative bacteria are recognized by TLR4, while TLR5 recognizes the bacterial protein flagellin. Nucleic acids serve as ligands for TLR3, 7, 8, and 9, and TLR2 is specific for lipoproteins.

The binding of a TLR ligand to the N-terminal ectodomain of a TLR prompts the formation of TLR homodimers or heterodimers. Following dimerization, the TLR signals are transduced via a cytoplasmic C-terminal Toll IL-1 receptor (TIR) domain to a set of adapter proteins.

Downstream, TLR signaling engages two distinct pathways, in which either TRIF (TICAM2) or My D88 serves as a key component. Both pathways culminate in the induction of inflammatory cytokines (TNF a, IL-6, IL-12), type I interferons (IFN-a, IFN-b), or apoptosis. Furthermore, TLR signaling induces dendritic cell maturation and consequently contributes to the adaptive immune response. Cytokine signaling is disturbed, permit-

ting an inflammatory reaction translated to an adverse drug reaction. In our patient, the shake protein induced very high levels of proinflammatory markers, indicating the role of inflammation in the adverse event [30].

In summary, the case reported here, which includes clinical, biochemical, and immunological data, implicates the combined role of the tea and shake protein in inducing hepatocytotoxicity in susceptible individuals. Since the number of active consumers of these products is large and rapidly increasing, it is important for physicians, as well as for consumers to be aware of the potential HILI resulting from consumption of these products. In our opinion, the main concern is that although these reactions are rare, there is a risk that the public and primary care medical personnel will fail to identify this problem early enough and stop the consumption of these products if there is any clinical suspicion of HILI or immune-mediated HILI (iHILI).

Drugs are prescribed worldwide to treat diseases, but they also carry the risk of idiosyncratic drug-induced liver injury (iDILI). Diagnosis of iDILI is made in a small number of cases, limiting clinical experience. To identify the mechanism of action, there is a need of collaborative work between clinicians and laboratories. We suggest that utilizing the LTA may enable the identification of more culprit drugs and elucidate possible mechanistic steps in the causality of iDILI and iHILI.

The main problem is to establish causality. The Roussel Uclaf Causality Assessment Method (RUCAM) is a diagnostic algorithm based on strong evidence to solve complex processes. The RUCAM is globally used to assess causality in around 100,000 published cases of iDILI worldwide. The RUCAM helps to establish a list of medicines implicated in iDILI and to describe clinical and mechanistic features of iDILI caused by various drugs [30–33].

In addition, the Liverpool and DILIN (Drug Induced Liver Injury Network) scores recently applied in iDILI cases of patients treated with immune checkpoint inhibitors (ICIs) [34,35]. In addition, herb-medication interactions might lead to hepatotoxicity [36,37]. LTA-based iDILI cases are helpful in elucidating pathogenetic steps, such as immune reactions and genetic predisposition. Our work achieves consistency in data collection, analysis, and the presentation of specific clinical, biochemical, immunological, and pathogenetic features.

**Author Contributions:** The patient was treated by S.M., A.A. and Y.M. at Kaplan Medical Center, Rehovot, Israel. All the immunology and toxicity assays were performed by M.G.N., Toronto, ON, Canada. M.G.N. and S.M. wrote the article. The clinical and routine biochemical analyses were performed at Kaplan Medical Center. All authors have read and agreed to the published version of the manuscript.

**Funding:** The laboratory research work was supported by In Vitro Drug Safety and Biotechnology.

**Institutional Review Board Statement:** The patient was treated at the Kaplan Medical Center. No IRB was needed.

**Informed Consent Statement:** The patient agreed to publish the data.

**Data Availability Statement:** Data are contained within the article.

**Conflicts of Interest:** The authors declare no conflicts of interest. All the authors contributed actively to the review. All authors disclose that they have no actual or potential conflicts of interest, including any financial, personal, or other relationships with other people or organizations that could inappropriately influence or be perceived to influence their work.

## Abbreviations

| | |
|---|---|
| ALT | Alanine aminotransferase (glutamic-pyruvic transaminase, GPT) |
| ALP | Alkaline phosphatase |
| AST | Aspartate amino-transferase (glutamic-oxalo-acetic transaminase, GOT) |
| AUC | Area under the curve |
| BCAA | Branched chain amino acids |
| CB | Conjugated (direct) bilirubin |
| CCL2 (MCP-1) | Metalloproteinase 1 |
| CCL5 (RANTES) | Regulated upon activation, normal T cell expressed and presumably secreted |
| CCR5 | Ligand (MIP)- macrophage inflammatory protein -1 |
| CIOMS | Council for International Organizations of Medical Sciences |
| DILI | Drug-induced liver injury |
| ECM | Extracellular matrix |
| ELISA | Enzyme-linked immunosorbent assay |
| GGT | $\gamma$-glutamyl-transferase ($\gamma$-glutamyl-transpeptidase, GGTP) |
| HILI | Herbal-induced liver injury |
| IFN$\beta$ | Interferon beta |
| IFN $\gamma$ | Interferon gamma |
| IL | Interleukin: IL1-$\alpha$ IL1$\beta$, IL2, IL4, IL6, IL8, IL 15 |
| INR | International Normalized Ratio |
| LTA | Lymphocyte toxicity assay |
| MMP | Matrix metalloprotease |
| MTT | 3-(4,5-dimethyl-thiazol-2-yl)-2,5-dio-phenyl tetrazolium bromide |
| NF$\kappa$B | Nuclear factor $\kappa$-B (transcription factor) |
| NKT | Natural killer cells |
| PDGF | Platelet-derived growth factor |
| RUCAM | Roussel Uclaf Causality Assessment Method |
| SDH | Succinate dehydrogenase |
| TGF | $\beta$ Transforming growth factor beta |
| TLR | Tall like receptor |
| TNF$\alpha$ | Tumor necrosis factor alpha |
| TRAIL | Tumor necrosis factor related apoptosis-inducing ligand-(APO-2L) |
| ULN | Upper limit of the normal reference range |
| VEGF | Vascular endothelial growth factor |
| Acute inflammation | elevated CCL5 (RANTES), IL1–$\alpha$, IL1–$\beta$, IL6, IL8, IL-15, TNF–$\alpha$ |

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
