# Peer review of "Phytotherapy-Induced Hepatocytotoxicity: A Case Report"

_cimb, doi:10.3390/cimb46070448_

Round 1
Reviewer 1 Report
Comments and Suggestions for Authors
The manuscript is interesting, in my view, it is worth publishing possible side effect of natural products, as long they might have also adverse effects. However, the organization of the manuscript needs a lot of improuvement. In this form is very difficult to follow. For example, there is a chapter entitle “Pathology” in my view the name that does not say too much… Than, no subchapters are present until almost the end of the chapter when a new subchapter entitled again pathology is introduced…Some information is repeated during the chapter, the figure needs a better graphics, with some explanatory legend… I think more data also may be showed as tables and figures… In my opinion the study needs the approval pf the institutional ethical comity, not only the consensus of the patient.
Comments on the Quality of English LanguageEnglish is generally ok, it is obvious that it is written by somebody with advanced English skills, however I would pay attention to some paragraphs that can have a more formal and clear English style.
Author Response
Thank you very much for reviewing our paper. We improved the presentation of material, methods and results as suggested by you. Dr. Malnick is a native English clinicians. He received his medical degree at Oxford University. He and I wrote and edited the English.
Reviewer 2 Report
Comments and Suggestions for Authors
This case report provides interesting and potentially important information on green tea-protein supplementation related toxicity. It is well performed and clearly reported. However, some minor improvements would do good to the paper:
1. Ideally, the composition of the supplements should be provided as analyzed by the authors - green tea calls for caffeine (and other xanthines) and polyphenol (such as ECGC, ECG and other green tea typical components) content, for instance using HPLC, and the protein amino-acid composition and content. It is one of the important limitations of the present version of the report.
Even more ideally, would be to perform a metabolomic analysis of the patient's blood sample (if still there) and attempting to interpret this. - e.g from the pint of view of altered amino acids metabolism or presence of green tea metabolites.
2. More information about the doses that were consumed by the patient, if available (at least an estimation).
3. In the discussion, some consideration should be added regarding the constituents of the supplement that may have caused the reported reaction. A role of caffeine or polyphenols from tea or some specific proteins/ peptides/amino acids could be discussed. Do you think it could be associated with oxidative stress (from polyphenols)? or rather there are other mechanisms?
4. The title should be more specific - now it sounds rather like a review paper - how about this:
"Phytotherapy-Induced Hepatocytotoxicity caused by a green tea - protein cocktail" or something in this style?
Author Response
1- Ideally, the composition of the supplements should be provided as analyzed by the authors - green tea calls for caffeine (and other xanthines) and polyphenol (such as ECGC, ECG and other green tea typical components) content, for instance using HPLC, and the protein amino-acid composition and content. It is one of the important limitations of the present version of the report.
Even more ideally, would be to perform a metabolomic analysis of the patient's blood sample (if still there) and attempting to interpret this. - e.g from the pint of view of altered amino acids metabolism or presence of green tea metabolites.
Response
We agree with the comment. However, we had the product and not the components of the tea or protein. As a result we analyzed the effect of tea, protein and their combination only.
2. More information about the doses that were consumed by the patient, if available (at least an estimation).
The patient consumed the products in the doses recommended by the product description.
3. In the discussion, some consideration should be added regarding the constituents of the supplement that may have caused the reported reaction. A role of caffeine or polyphenols from tea or some specific proteins/ peptides/amino acids could be discussed. Do you think it could be associated with oxidative stress (from polyphenols)? or rather there are other mechanisms?
Thank you for reviewing the paper.
1-We can't speculate on the oxidative stress. The individual is a healthy, young person. He loves sport, he does not drink alcohol and dose not take medication. His daily routine includes exercise.
To perform a metabolomic analysis of the patient's blood metabolism is a great idea. Thank you.
2. More information about the doses that were consumed by the patient, if available (at least an estimation).
He followed the instruction given by the company.
3. In the discussion, some consideration should be added regarding the constituents of the supplement that may have caused the reported reaction. A role of caffeine or polyphenols from tea or some specific proteins/ peptides/amino acids could be discussed. Do you think it could be associated with oxidative stress (from polyphenols)? or rather there are other mechanisms
4. The title should be more specific - now it sounds rather like a review paper - how about this:
"Phytotherapy-Induced Hepatocytotoxicity caused by a green tea - protein cocktail" or something in this style?
Thank you for the suggestion.
Reviewer 3 Report
Comments and Suggestions for Authors
Abstract: it is incomprehensible, the background did not establish the context of the report to understand the main purpose. The English editing language needs to be seriously improved because some sentences make no sense. Some abbreviations mentioned in the abstract have wrong definitions.
Introduction: It is practically impossible to read and understand the scientific logic that supports this work. There is a succession of sentences without any connection between them that helps for the comprehension. For example, you introduced by defining DILI and hepatotoxicity, then suddenly started to talk about type I and II lesions but what are they supposed to classify?
What is the difference between drug-induced liver injury and drug-induced hepatotoxicity?
Case report: There is the same problem concerning the English editing language. Except for that, the report of an isolated man combined with his own physiological conditions can be considered as a first alarming sign but does not seem justified as case reports of the hepatotoxicity of green tea or protein powder.
Pathology: Based on in vitro studies of lymphocyte culture, It seems more interesting. the paragraph (lines 136-143) was repeated twice.
Discussion: In this part, the author presented practically the same information stated in the previous part without real elements confirming or justifying their results with the references from previous studies associated. thus, it seems that the study has been previously discussed and then this part does not have any specific role in the manuscript despite the interested ideas developed.
Comments on the Quality of English LanguageTHe English editing language is practically incomprehensible and needs to be improved. Missing of punctuation or inappropriately used, missing of connections between sentences, there is too much mix of verbs tense that have to be uniformized to ease the lecture.
Author Response
Thank you for reviewing the manuscript.
1- The first author, Dr. Malnick is a native English clinician that finish his medical school at Oxford University, England.
All the 3 clinicians are hepatologists diagnosing HILI and DILI. I, M. Neuman, I am a long time researcher on the same subject and a clinical chemist certified by the Canadian Royal Academy of Clinical Chemist to give laboratory diagnosis.
We took the constructive suggestions and incorporated in the text.
Thanks for the review,
Reviewer 4 Report
Comments and Suggestions for Authors
This case report present clinical and laboratory evidences implicating hepatotoxicity induced by intake of protein powder containing green tea and branched chain amino acids. Although this case report could be useful especially for the clinicians, major improvements have to be done in order to make this paper suitable for publication. Abstract should be completely rewritten in order to better reflect the presented results. Authors should rewrite the paper in order to decrease the high percentage of overlay with published data. In addition, in discussion part references are missing and table 2 is placed after the conclusion. Specification of product is missing.
Comments on the Quality of English LanguageModerate improvements are noted.
Author Response
Our case report present clinical and laboratory evidences implicating hepatotoxicity induced by intake of protein powder containing green tea and branched chain amino acids.
"Although this case report could be useful especially for the clinicians, major improvements have to be done in order to make this paper suitable for publication.
Abstract should be completely rewritten in order to better reflect the presented results. Authors should rewrite the paper in order to decrease the high percentage of overlay with published data. In addition, in discussion part references are missing and table 2 is placed after the conclusion. Specification of product is missing."
Response
We made the necessary changes.
Thank you for suggestions.
Reviewer 5 Report
Comments and Suggestions for Authors
the research topic is very interesting and the authors focus on investigating how herbal medicine influences on liver health. generally in this case report the authors try to investigate whether the consumption of protein supplements and green tea produce liver diseases, for that purpose in vitro study was perform
the introduction is concise gives all the necessary information about previous researchers and focuses on the most important facts.
the methodology is described well with sufficient information for reader to repeat the experiments.
the presented results were analyzed and discussed properly pointing out that hepatotoxicity in this case was the sum of protein and green tea toxicity and correlated with sensitivity of the patient to it.
the discussion is well-organized and adequate for research but there is few point for improvement, for example, better explanation of The Naranjo score and Hy laws) their importance and how to interpret their values)
what abbreviations CIOMS means?
are there any other parameters that decreased or increased during herbs-induced hepatotoxicity?
also I suggests authors to pointed out that after discontinuation of supplements conception the symptoms of liver disease slowly diminished.
the conclusion is supported by the results underlining the most important findings.
Comments on the Quality of English Languagethe paper is easy to read and understand hence English style and grammar are satisfactory.
Author Response
Dear reviewer,
We appreciate very much your words. We responded in the text to the queries regarding the different drug-induced liver injury (DILI) scores. We hope that the explanation is satisfactory.
Round 2
Reviewer 1 Report
Comments and Suggestions for Authors
Dear Author,
I found a significative improvement in both English, style and content. In my view the manuscript can be published in this version, if ethical approval is not mandatory according to national legislation.
Reviewer 4 Report
Comments and Suggestions for Authors
The authors improved the paper, however overall quality is still not proper for publication. Lot of mistakes have been made during the revision process. Please take into account the comments given below:
Line34-36: The sentence should be rewritten.
Line 86, 89: Check the font
Line 133: The sentence is duplicated and should be deleted.
Table 2 is not proper. Authors should first mention in the text the results presented in table 2.
Text given in lines 295-297 should be placed before Figure 1. Moreover, new inserted sentences in lines 294, 298-299 are not properly placed.
Line 305-306: ?
Line 334, (EVL Nutrition), bracket is missing
Line 339: “The CIOMS score is the scoring system of choice” should be deleted
Line 347-350 these sentences after insertation are not correct. They should be rewritten.
Line 387: “the assessment of clinical assessment” Sentence should be rewritten
Comments on the Quality of English Language
Minor improvements are needed.